# 1,4-Naphthoquinone Motif in the Synthesis of New Thiopyrano[2,3-*d*]thiazoles as Potential Biologically Active Compounds

**DOI:** 10.3390/molecules27217575

**Published:** 2022-11-04

**Authors:** Andrii Lozynskyi, Julia Senkiv, Iryna Ivasechko, Nataliya Finiuk, Olga Klyuchivska, Nataliya Kashchak, Danylo Lesyk, Andriy Karkhut, Svyatoslav Polovkovych, Oksana Levytska, Olexandr Karpenko, Assyl Boshkayeva, Galiya Sayakova, Andrzej Gzella, Rostyslav Stoika, Roman Lesyk

**Affiliations:** 1Department of Pharmaceutical, Organic and Bioorganic Chemistry, Danylo Halytsky Lviv National Medical University, Pekarska 69, 79010 Lviv, Ukraine; 2Institute of Cell Biology of National Academy of Sciences of Ukraine, Drahomanov14/16, 79005 Lviv, Ukraine; 3Department of Technology of Biologically Active Substances, Pharmacy and Biotechnology, Lviv Polytechnic National University, Bandera 12, 79013 Lviv, Ukraine; 4Department of Organization and Economics of Pharmacy, Danylo Halytsky Lviv National Medical University, Pekarska 69, 79010 Lviv, Ukraine; 5Enamine Ltd., 23 Alexandra Matrosova, 01103 Kyiv, Ukraine; 6Department of Pharmaceutical and Toxicological Chemistry, Pharmacognosy and Botany, Asfendiyarov Kazakh National Medical University, Almaty 050000, Kazakhstan; 7Department of Organic Chemistry, Poznan University of Medical Sciences, Grunwaldzka 6, 60-780 Poznan, Poland; 8Department of Biotechnology and Cell Biology, Medical College, University of Information Technology and Management in Rzeszow, Sucharskiego 2, 35-225 Rzeszow, Poland

**Keywords:** 1,4-naphthoquinones, *hetero*-Diels-Alder reaction, thiopyrano[2,3-*d*]thiazoles, X-ray, anticancer activity, cytotoxicity, DNA intercalation

## Abstract

A series of 11-substituted 3,5,10,11-tetrahydro-2*H*-benzo[6,7]thiochromeno[2,3-*d*][1,3]thiazole-2,5,10-triones were obtained via *hetero*-Diels-Alder reaction of 5-alkyl/arylallylidene/-4-thioxo-2-thiazolidinones and 1,4-naphthoquinones. The structures of newly synthesized compounds were established by spectral data and a single-crystal X-ray diffraction analysis. According to U.S. NCI protocols, compounds **3.5** and **3.6** were screened for their anticancer activity; 11-Phenethyl-3,11-dihydro-2*H*-benzo[6,7]thiochromeno[2,3-*d*]thiazole-2,5,10-trione (**3.6**) showed pronounced cytotoxic effect on leukemia (Jurkat, THP-1), epidermoid (KB3-1, KBC-1), and colon (HCT116wt, HCT116 p53-/-) cell lines. The cytotoxic action of **3.6** on p53-deficient colon carcinoma cells was two times weaker than on HCT116wt, and it may be an interesting feature of the mechanism action.

## 1. Introduction

1,4-Naphthoquinones are a large group of biologically active molecules found in various higher plants (plant families *Juglandaceae*, *Plumbaginaceae*, *Boraginaceae*, *Ebenaceae*, *Droseraceae*, *Lythraceae*, *Rubiaceae*, *Balsaminaceae*, *Bignoniaceae*, and *Ulmaceae*), lichens, bacteria, and invertebrates (especially in arthropods and echinoderms) [1]. In natural sources, 1,4-naphthoquinones often exist in a reduced form or as glycosides [2]. Some drugs, products used in cosmetology, and biologically active additives contain naphthoquinone derivatives as an active component. In particular, vitamin K_1_ and menadione, as their synthetic analogs, are well-known drugs used as hemostatic agents, causing an increase in the synthesis of prothrombin and proconvertin [3,4]. Atovaquone is a synthetic antiprotozoal agent for the prevention and treatment of *Pneumocystis jirovecii* pneumonia (PCP) and malaria [5,6]. Menatetrenone is a form of vitamin K_2_ for the treatment of osteoporosis to stimulate osteogenesis. Currently, several 1,4-naphthoquinones (i.e., phylloquinone for regulation of blood coagulation, bone metabolism, and vascular biology), lawsone (natural dye), naphthazarin (natural dye), and atovaquone (antineumococcal), are used both parenterally and externally [7].

In general, 1,4-naphthoquinone derivatives, similarly to other quinones, manifest their pharmacological potential via two mechanisms [1]. The first is related to the redox properties of naphthoquinones, which are easily reduced and re-oxidized under physiological conditions. Thus, naphthoquinone derivatives in the presence of molecular oxygen and appropriate reducing agents catalyze the transfer of electrons from NADPH or thiols, which leads to the generation of various reactive oxygen species (ROS); in particular, superoxide anion, hydroxyl radicals, and hydrogen peroxide [8,9]. The second mechanism is based on the high electrophilicity of 1,4-naphthoquinone derivatives. This property allows 1,4-naphthoquinones to form covalent bonds with nucleophilic agents and interact with thiol groups of proteins, glutathione, and nucleophilic amino acid groups; for example, with the terminal amino group of lysine [1]. Accordingly, this complex of properties made it possible to identify several highly active anticancer [10,11,12], antibacterial [13], antifungal [14], anti-inflammatory [15,16], and antiparasitic agents [17,18,19,20] (Figure 1). Furthermore, some 1,4-naphthoquinone derivatives were found as efficient inhibitors of proteasomes [21], N-acetyltransferase [11], cyclin-cyclin-dependent kinase [22], aldose reductase [16], topoisomerases I and II [23,24], heat shock proteins [25], DNA gyrase [26], phosphatidylinositol 3-kinase [27], and inhibitors of Stat3 [28] and cancer stem cell cascades [11].

Additionally, 1,4-naphthoquinone derivatives are an interesting scaffold for various types of chemical transformations. Thus, being active dienophiles, they undergo Diels-Alder reactions, which are often accompanied by variability in regio- and diastereoselectivity processes depending on the selected heterodienes and substituents in the naphthoquinone fragment providing various types of polycyclic systems [29,30,31,32,33,34] (Figure 2). It is also worth noting that Diels-Alder reactions of naphthoquinone derivatives as dienophiles make it possible to obtain some biologically active natural quinones, particularly angucyclinone antibiotics, which undergo stages of clinical trials [35]. Accordingly, the purpose of this work was to apply 1,4-naphthoquinone as the dienophile in *hetero*-Diels–Alder reactions for the synthesis of novel thiopyrano[2,3-*d*]thiazole derivatives. It is worth noting that thiopyrano[2,3-*d*]thiazole derivatives have shown a broad range of biological activities, such as anticancer [36,37,38], antioxidant [39], antitrypanosomal [40], anti-inflammatory [41], etc. The synthesized compounds were also evaluated for their primary anticancer activity, cytotoxicity, and DNA intercalation in vitro.

## 2. Results

### 2.1. Chemistry

The starting 5-arylallylidene- and 5-(cyclo)alkylidene-4-thioxo-2-thiazolidinones, as heterodienes in the synthesis of target thiopyranothiazole derivatives, were synthesized via the Knoevenagel condensation of 4-thioxo-2-thiazolidinone (isorhodanine) and appropriate aldehydes or ketones using ethylenediammonium diacetate (EDDA) as the catalyst in ethanol medium. The *hetero*-Diels–Alder reaction was carried out in boiling acetic acid in the presence of a catalytic amount of hydroquinone as a side polymerization inhibitor, and target 11-substituted-3,5,10,11-tetrahydro-2*H*-benzo[6,7]thiochromeno[2,3-*d*][1,3]thiazole-2,5,10-triones **3.1**–**3.6** were obtained with good yields (Figure 1). It should be noted that aliphatic aldehydes with isorhodanine react less actively with low yields, and the resulting 5-alkylidene-4-thioxo-2-thiazolidinones are quite difficult to isolate [42]. Accordingly, a three-component reaction of isorhodanine, aliphatic aldehyde, and dienophile was used to synthesize target thiopyrano[2,3-*d*]thiazoles. The reaction of isorhodanine, 3-phenylpropionaldehyde, and 1,4-naphthoquinone in acetonitrile in the presence of EDDA as catalyst afforded pure target derivative **3.6**. It is worth noting that during the *hetero*-Diels-Alder reaction, the products of [4+2]-cycloaddition undergo spontaneous oxidation (dehydrogenation) by excess naphthoquinone, which has been reported previously [38].

The structures of the synthesized compounds were elucidated by spectral data. Thus, protons of the naphthoquinone moiety in the ^1^H NMR spectra of synthesized thiopyranoids showed characteristic signals at δ ∼ 7.02–8.49 ppm. The signal of the CH proton in the C-16 position appeared as a singlet or multiplet at 4.42–6.77 ppm. Protons attributed to the phenylpropionyl residue of **3.6** showed two multiplets at δ 1.94–2.62 ppm. Low field ^1^H NMR spectra gave signals in a range of 10.80–11.89 ppm, which were assigned to the signal of the amide proton. In the ^13^C NMR spectra of the synthesized compounds, the signals observed at δ 166.9–185.3 were assigned to the carbonyl group (C=O) of the naphthoquinone fragment (Appendix A).

The structure of the synthesized 11-phenethyl-3,11-dihydro-2*H*-benzo[6,7]thiochromeno[2,3-*d*]thiazole-2,5,10-trione (**3.6**) was confirmed by X-ray diffraction analysis. The molecular structure and the atom-labeling Scheme are illustrated in Figure 3.

The compound has a rigid tetracyclic system of 3,11-dihydro-2*H*-benzo[6,7]thiochromeno[2,3-*d*]thiazole-2,5,10-trione with the planar or approximately planar thiazolin-2-one and naphthalene-1,4-dione ring moieties (r.m.s. deviation 0.0041 and 0.0209 Å, respectively), and the slightly puckered 4*H*-thiopyran ring (r.m.s. deviation 0.0846 Å).

Within the thiazolin-2-one system, the presence of a secondary amide group was noted. The position of the hydrogen atom bound to the N-3 atom was determined from the difference Fourier map and freely refined. Its location in the mentioned position was confirmed by the hydrogen bond N3–H3⋯O18^i^ [*Donor*–H: 0.89(4) Å, H⋯*Acceptor*: 1.89(4) Å, *Donor*⋯*Acceptor*: 2.744(4) Å, *Donor*–H⋯*Acceptor*: 160(4)°], in which the carbonyl oxygen atom plays the role of the proton acceptor.

Our studies have shown that in the 3,11-dihydro-2*H*-benzo[6,7]-thiochromeno[2,3-*d*]thiazole-2,5,10-trione ring system there are double bonds between the nodal C-4/C-17 and C-6/C-15 atoms. The found interatomic distances C4–C17 [1.339 (4) Å] and C6–C15 [1.349(4) Å] are close to the literature length of the double C–C bond [1.331(1) Å] [43].

Structural investigations have shown that in the crystal, the phenyl ring of the phenethyl residue at the stereogenic atom C-16 occupies two alternative positions, marked *a* and *b*, as a result of swing motion and simultaneous rotation of the ring around the C23–C26 axis. The angle C23*a*−C22−C23*b* and the dihedral angle Ph_*a*−Ph_*b* found are 17.2(4) and 26.9(3)°, respectively. Occupancy factors for alternative positions *a* and *b* of the phenyl ring are 0.5.

The arrangement of the phenethyl residue is described with torsion angles C4–C17–C16–C21, C17–C16–C21–C22, and C16–C21–C22–C23a/23*b*, which reveal values of 104.1(3), −67.7(3), and −170.3(4)/171.9(4)°. Moreover, the alternative phenyl rings *a* and *b* belonging to phenethyl moiety form with the mean plane of 4*H*-thiopyran ring the dihedral angles of 68.8(2) and 60.8(2)°, respectively.

### 2.2. Biological Evaluation

#### 2.2.1. Cytotoxicity Activity Screening

Considering the results of previous studies of fused thiazolidinones and their analogs, a series of thiopyrano[2,3-*d*]thiazoles with naphthoquinone fragments in the structure were studied for their anticancer activity. Thus, synthesized thiopyranothiazole derivatives **3.5** and **3.6** were selected by the National Cancer Institute (NCI), U.S., for their anticancer activity at 10 μM concentration toward a panel of sixty cancer cell lines representing nine different types (leukemia, melanoma, lung, colon, CNS, ovarian, renal, prostate, and breast cancers) (Appendix A). Selection for screening based on new derivatives’ ability to add diversity to the NCI small molecules collection and anticancer assays were performed according to the NCI guidelines and protocols previously described [44,45,46]. The compounds were added at the mentioned concentration, and the cell cultures were incubated for 48 h. The results for each compound were reported as the growth percent (GP%) of treated cells compared with untreated control cells. The screening results are shown in Table 1.

The studied naphthoquinone-substituted thiopyrano[2,3-*d*]thiazoles demonstrated inhibition of tested cancer cell lines growth in the in vitro screening. The GP of breast cancer T-47D cells was 51.21% under treatment with compound **3.5**, and the GP of melanoma MALME-3M cells was 45.25% under treatment with compound **3.6**.

We continued the cytotoxicity study of synthesized derivatives **3.1**–**3.6** toward tumor and pseudo-normal cells in vitro. The MTT cell viability assay was performed 72 h after cells treatment with various concentrations of studied compounds and doxorubicin, a reference drug. The cell viability and the IC_50_ values are shown in Figure 4 and Table 2. The most active was compound **3.6**, and leukemic cell lines were the most sensitive to its action. It was cytotoxic to Jurkat T-leukemia cells at all tested concentrations with the half-maximal inhibitory concentration (IC_50_) of 0.76 µM. The THP-1 cells, monocytes isolated from peripheral blood from an acute monocytic leukemia patient [47], were also sensitive to the **3.6** treatment. The IC_50_ of this compound was 7.94 µM. The **3.6** inhibited the viability of epidermoid carcinoma (KB3-1 and its ABCB1-overexpressing subline KBC-1) and colon carcinoma (HCT116wt and its p53 knockdown subline HCT116 p53-/-) cells. It is known that ABCB1 (P-glycoprotein), MRP1/ABCC1 (multidrug resistance protein 1), and BCRP/BCG2 (breast cancer resistance protein) have been reported to be key players in resistance to chemotherapy [48]. It should be noted that the **3.6** demonstrated a pronounced growth inhibition effect on KBC-1 (IC_50_ = 12.81 µM). The IC_50_ of **3.6** was 27.66 µM for KB3-1 cells. We found significantly lower IC_50_ values of **3.2**, **3.4**, and **3.6** for HCT116wt compared with those for the HCT116 p53-/- cell line. The IC_50_ ranged from 5.54 to 6.81µM and from 12.34 to >50 µM, respectively. Thus, the p53 status of colon carcinoma cells influenced the anti-tumor effects of the studied 3,5,10,11-tetrahydro-2*H*-benzo[6,7]thiochromeno[2,3-*d*][1,3]thiazole-2,5,10-trione. The p53 has diverse mutations in almost all human tumors, stimulating their hyper-proliferation, invasion/metastasis, and thus, influencing the potency of various chemotherapeutics (i.e., platins and anti-metabolites). The p53-deficient or p53-mutant tumors often possessed a more aggressive phenotype and more pronounced chemo- and radio-resistance [49]. Based on our findings, one can assume a p53-dependent mode of action for **3.6** toward colon cancer cells.

Derivatives **3.1**, **3.2**, and **3.6** possessed an anti-proliferative effect on MCF-7 (hormone-dependent, estrogen, and progesterone receptor positive) cells with a similar IC_50_ of 9.19, 8.47, and 8.94 µM, respectively [50]. K562 cells were sensitive to compounds **3.3** and **3.4**, with an IC_50_ of 13.00 and 7.11 µM, respectively. Doxorubicin was more cytotoxic for epidermoid, colon, breast carcinoma, and melanoma cells (Figure 4, Table 2). Compound **3.5** showed weak activity on cell lines used in our work. A reference compound was also used, 1,4-naphthoquinone (1,4-NQ), with weak activity towards KB3-1 and KBC-1 cell lines (IC_50_ 20.74 and 8.33 µM, respectively).

The **3.6** showed low toxicity in human keratinocytes of the HaCaT line. The IC_50_ of **3.6** was >100 µM for HaCaT cells. The isolated normal human peripheral blood lymphocytes and murine macrophages of the J774.2 line were more sensitive to **3.6** treatment. It induced a 50% reduction in the viability of isolated normal human lymphocytes at 58.66 µM. The **3.6** reached the IC_50_ value at 9.57 µM for the J774.2 macrophage cell line. The **3.6** derivative caused a moderate reduction in the viability of HaCaT, J774.2 cell lines, and isolated normal human lymphocytes. The blood-derived cells were more sensitive to the **3.6** treatment. Doxorubicin reduced the survival of pseudo-normal cells and isolated normal lymphocytes at the IC_50_ value of 0.8–1.0 µM (Figure 5, Table 2).

Compound **3.6** possessed a high anti-proliferative effect on selected tumor cells and moderate toxicity on pseudo-normal ones. Thus, it was chosen for further experimental research in vitro.

#### 2.2.2. Reactivity with Reduced Glutathione (GSH)

The reactivity of unmetabolized compound **3.6** was assessed in the test with the model soft nucleophile reduced glutathione (GSH). It has been found that after incubation with **3.6**, the level of GSH decreases and does not increase with adding sodium borohydride, which suggests the formation of covalent GS-adducts that are not reduced to GSH with sodium borohydride, unlike oxidized glutathione GSSG (Figure 6).

#### 2.2.3. Cellular Morphology of KB3-1 Cells Induced by Compound **3.6**

To elucidate the primary death mechanisms in the treated cells with the **3.6** derivative, we assessed apoptosis by DNA laddering assay, fluorescent microscopy after cell staining with Hoechst-33342, DNA interacting spectroscopic, and DNA/methyl green replacement assays.

The **3.6** caused significant cytomorphological alterations in KB3-1 cells, which were found to be shrunk, with condensed chromatin and membrane blabbing (Figure 7B) compared to the control (Figure 7A). One can also see giant KB3-1 cells with abnormal nuclei and looser chromatin (Figure 7B). The mitotic catastrophe may occur during or after aberrant mitosis. Mitotic catastrophe has been reported as a special example of apoptosis affecting mitochondrial membrane permeabilization and caspase activation [51]. The control KB3-1 cells exhibited properly shaped intact nuclei. Doxorubicin caused chromatin condensation and membrane blabbing (Figure 7C). Thus, compound **3.6** induced pro-apoptotic cytomorphological changes in treated KB3-1 cells. As shown in Figure 7B, compound **3.6** was able to red fluorescence (DIC + Red channel) in the cells, similar to doxorubicin (Figure 7C), and was more concentrated in the nucleus area.

#### 2.2.4. DNA Laddering under Treatment of **3.6**

In the presence of compound **3.6**, we did not detect a typical apoptotic laddering in Jurkat cells (Figure 8). One can see that compound **3.6** at 1 µM induced slight laddering of DNA. The **3.6** at 2.5 µM and 5 µM induced necrotic degradation of DNA. We assumed that the **3.6** at 10 µM and 25 µM induced extreme DNA fragmentation that could not be seen on the gel.

Compound **3.6** induced both apoptotic and necrotic death of KB3-1 cells. Doxorubicin at 0.5 µM induced more necrotic changes in treated Jurkat cells than apoptotic ones (Figure 8).

#### 2.2.5. DNA Interacting Ability of Compound **3.6**

The method investigates conformational changes in the DNA; for example, when DNA is exposed to an intercalating, alkylation, or other classes of DNA-binding agents. It is based on the oxidative reaction of potassium permanganate with pyrimidine bases. Compounds that interact with DNA distort its duplex structure, thus exposing pyrimidine bases for oxidation by KMnO_4_, which generates products that can be detected by spectrophotometry [52]. Different classes of DNA-binding compounds can be studied in such a way. Data obtained from samples with DNA incubated with the tested compound **3.6** showed a strong time and concentration-dependent increase in the oxidation level compared with control DNA (without the studied compound). Net A405 ranged from 0.00 to 0.50 in the presence of the compound, and Net A405 ranged from 0.02 to 0.045 in the control case (Figure 9). The obtained results indicated that compound **3.6** interacts in some way with DNA.

#### 2.2.6. DNA/Methyl Green Replacement Assay

In addition, we used DNA/methyl green colorimetric assay to study the possible interaction of compound **3.6** in DNA. Methyl green is found to be a DNA major-groove binding compound [53] and reversibly binds polymerized DNA. This assay was used to measure the displacement of methyl green from DNA by compounds with the ability to intercalate DNA [54]. Tested compound **3.6** could intercalate between two complementary base pairs in double-stranded DNA, and, dependent on concentration, the percentage of methyl green replacement ranged from 35.00 to 39.64% (Figure 10). Indeed, 1,4-NQ showed a stronger ability to methyl green replacement; in concentration 1µM it displaced 65% of methyl green. Doxorubicin, which was used as a positive control, in concentration 1µM, had a similar effect to compound **3.6**, but in concentration 10 µM replaced methyl green as being two times more effective. Together with data obtained in through spectroscopic assay, and red fluorescence in the nucleus area (morphology data), this result indicated that one of the possible mechanisms of action of compound **3.6** is DNA intercalation.

## 3. Materials and Methods

### 3.1. General Information

All materials were purchased from commercial sources and used without purification. Melting points were measured in open capillary tubes and were uncorrected. The elemental analyses were performed using a Thermo Scientific FlashSmart Elemental Analyzer. The ^1^H and ^13^C NMR spectra were recorded on a Varian Gemini (^1^H at 400 and ^13^C at 100 MHz) instrument in DMSO-*d_6_*. Chemical shifts (δ) were given in ppm units relative to tetramethylsilane as reference (0.00). The purity of all obtained compounds was checked by TLC on Silufol-254 plates (Eluent EtOAc/ Benzene 1:2). The starting 5-ylidene-4-thioxo-2-thiazolidinones 2.1–2.5 were obtained according to the method previously described [41,55].

Human colon carcinoma HCT116 cells, human breast adenocarcinoma cells of MCF-7 line, human T-leukemia Jurkat cells, human chronic myelogenous leukemia K562 cells, and human keratinocytes of HaCaT line were from the Cell Collection of R.E. Kavetsky Institute of Experimental Pathology, Oncology and Radiobiology (Kyiv, Ukraine). Murine macrophages of J774.2 line were a generous gift from Professor Sir John Vane (William Harvey Research Institute, London, UK) via Professor Janusz Marcinkiewicz (Jagiellonian University Medical College, Krakow, Poland). Human colon carcinoma HCT116 p53-/- cells with knockdown of P53 gene, as well as human epidermoid cervix carcinoma KB3-1 cells and its ABCB1-overexpressing subline KBC-1, were kindly provided by Professor W. Berger (Institute of Cancer Research, Medical University of Vienna, Austria). The phenotype of this cell line was stable, as periodically determined by Western blot analysis. Human leukemia monocytic THP-1 cells were kindly provided by Professor M. Herrmann (Department of Internal Medicine, Institute for Clinical Immunology and Rheumatology, University of Erlangen-Nuremberg, Germany). Cells were cultured in DMEM or RPMI-1640 medium supplemented with 10% fetal bovine serum (all were purchased from Biowest, Nuaille, France) at 37 °C in a humidified atmosphere containing 5% CO_2_.

### 3.2. Synthesis of 3,11-Dihydro-2H-benzo[6,7]thiochromeno[2,3-d]thiazole-2,5,10-triones ***3.1***–***3.5***

A mixture of appropriate 5-alkyl/arylallylidene/-4-thioxo-2-thiazolidinone (10 mmol) and 1,4-naphthoquinone (20 mmol) was refluxed for 1 h with a catalytic amount of hydroquinone (2–3 mg) in glacial acetic acid (10 mL), and then left overnight at room temperature. The precipitated crystals were filtered off, washed with methanol (5–10 mL), and recrystallized from the appropriate solvent.

*11-Styryl-3,11-dihydro-2H-benzo* [6,7]*thiochromeno [2,3-d]thiazole-2,5,10-trione* (**3.1**). Yield 70%, mp 338–340 °C (DMF:EtOH). ^1^H NMR (400 MHz, DMSO-d_6_): δ 6.77 (s, 1H, CH), 7.53–7.63 (m, 6H, CH, arom.), 7.69 (t, 1H, J = 7.8 Hz, arom.), 7.73 (t, 1H, J = 8.2 Hz, arom.), 8.12 (d, 1H, J = 16.4 Hz, CH), 8.24 (d, 1H, J = 8.4 Hz, arom.), 8.39 (d, 1H, J = 8.4 Hz, arom.), 11.04 (s, 1H, NH). ^13^C NMR (100 MHz, DMSO-d6): δ 36.2, 95.8, 118.7, 119.6, 122.4, 123.5, 126.6, 127.7, 128.1, 129.1, 129.2, 129.9, 131.4, 131.8, 147.8, 156.2, 175.2, 177.2, 179.7 ESI-MS *m*/*z* 404 (M + H)^+^. Anal.Calcd for C_22_H_13_NO_3_S_2_: C, 65.49; H, 3.25; N, 3.47. Found: C, 65.64; H, 3.08; N, 3.63.

*11-(2-Nitrostyryl)-3,11-dihydro-2H-benzo*[6,7]*thiochromeno[2,3-d]thiazole-2,5,10-trione* (**3.2**). Yield 69%, mp 330–332 °C (DMF:EtOH). ^1^H NMR (400 MHz, DMSO-d6): δ 6.02–6.08 (m, 1H, CH), 7.66 (t, 2H, J = 8.4 Hz, arom.), 7.74–7.77 (m, 2H, arom.), 7.89–7.93 (m, 1H, arom.), 8.10 (dd, 2H, J = 8.2,16.0 Hz, 2CH), 8.18 (d, 1H, J = 8.4 Hz, arom.), 8.29 (d, 2H, J = 8.2 Hz, arom.), 11.83 (s, 1H, NH). ^13^C NMR (100 MHz, DMSO-d6): δ 31.6, 98.1, 114.4, 118.8, 122.6, 125.1, 130.3, 132.6, 134.4, 134.9, 137.1, 139.1, 141.9, 144.2, 146.6, 149.7, 173.5, 174.9, 178.5. ESI-MS *m*/*z* 449 (M + H)^+^. Anal.Calcd for C_22_H_12_N_2_O_5_S_2_: C, 58.92; H, 2.70; N, 6.25. Found: C, 59.07; H, 2.82; N, 6.12.

*11-(2-Phenylprop-1-en-1-yl)-3,11-dihydro-2H-benzo*[6,7]*thiochromeno[2,3-d]thiazole-2,5,10-trione* (**3.3**). Yield 80%, mp > 350 °C (DMF:EtOH). ^1^H NMR (400 MHz, DMSO-d6): δ 0.96 (s, 3H, CH_3_), 5.77 (s, 1H, CH), 7.41–7.44 (m, 1H, arom.), 7.50–7.56 (m, 2H, CH, arom.), 7.60–7.65 (m, 1H, arom.), 7.69–7.73 (m, 3H, arom.), 7.79–7.82 (m, 1H, arom.), 8.28–8.32 (m, 1H, arom.), 8.46–8.49 (m, 1H, arom.), 10.80 (s, 1H, NH). ^13^C NMR (100 MHz, DMSO-d6): δ 17.1, 29.2, 86.1, 108.6, 122.4, 122.6, 125.4, 125.7, 126.2, 126.3, 126.4, 127.4, 128.0, 128.5, 144.4, 146.7, 164.2, 166.9, 177.6. ESI-MS *m*/*z* 418 (M + H)^+^. Anal.Calcd for C_23_H_15_NO_3_S_2_: C, 66.17; H, 3.62; N, 3.35. Found: C, 66.27; H, 3.40; N, 3.49.

*11,11-Dimethyl-3,11-dihydro-2H-benzo*[6,7]*thiochromeno[2,3-d]thiazole-2,5,10-trione* (**3.4**). Yield 81%, mp 230–232 °C (DMF:EtOH). ^1^H NMR (400 MHz, DMSO-d6): δ 1.73 (s, 3H, CH_3_), 1.89 (s, 3H, CH_3_), 7.82 (t, 1H, J = 7.6 Hz, arom.), 7.88 (d, 1H, J = 7.3 Hz, arom.), 7.94–8.02 (m, 2H, arom.), 11.65 (s, 1H, NH). ^13^C NMR (100 MHz, DMSO-d6): δ 21.0, 29.7, 106.3, 114.6, 125.8, 126.9, 135.1, 142.8, 146.2, 170.0, 172.0, 180.7. ESI-MS *m*/*z* 330 (M + H)^+^. Anal.Calcd for C_22_H_15_NO_3_S_2_: C, 65.17; H, 3.73; N, 3.45. Found: C, 65.24; H, **3.6**1; N, **3.6**2.

*Spiro[benzo*[6,7]*thiochromeno[2,3-d]thiazole-11,1’-cyclopentane]-2,5,10(3H)-trione* (**3.5**). Yield 72%, mp 231–233 °C (AcOH). ^1^H NMR (400 MHz, DMSO-d6): δ 2.14 (br.s, 2H, CH_2_), 2.39 (br.s, 2H, CH_2_), 2.81 (m, 4H, 2*CH_2_), 7.84 (m, 2H, arom.), 8.05 (m, 2H, arom.), 11.70 (s, 1H, NH). ^13^C NMR (100 MHz, DMSO-d6): δ 21.7, 31.2, 45.5, 106.5, 122.3, 127.8, 134.6, 136.3, 138.3, 143.3, 150.3, 176.0, 177.7, 185.3. ESI-MS *m*/*z* 356 (M + H)^+^. Anal.Calcd for C_16_H_11_NO_3_S_2_: C, 58.34; H, 3.37; N, 4.25. Found: C, 58.21; H, 3.52; N, 4.16.

### 3.3. Synthesis of 11-Phenethyl-3,11-dihydro-2H-benzo[6,7]thiochromeno[2,3-d]thiazole-2,5,10-trione ***3.6***

A mixture of isorhodanine (5 mmol), phenylpropionaldehyde (5.5 mmol), and 1,4-naphthoquinone (10 mmol) was heated at reflux for 2 h in MeCN (10 mL) in the presence of the catalytic amount of ethylenediaminediacetic acid. After cooling, the precipitate was filtered off, washed, and recrystallized from the appropriate solvent. Yield 70%, mp 200–202 °C (DMF:EtOH). ^1^H NMR (400 MHz, DMSO-d6): δ 1.94 (m, 2H, CH_2_), 2.62 (m, 2H, CH_2_), 4.42 (m, 1H, CH), 7.02 (m, 1H, arom.), 7.07–7.14 (m, 4H, arom.), 7.78–7.93 (m, 2H, arom.), 7.94–8.06 (m, 2H, arom.), 11.89 (s, 1H, NH). ^13^C NMR (100 MHz, DMSO-d6): δ 31.6, 34.3, 36.5, 107.0, 117.1, 126.2, 126.5, 127.1, 128.5, 128.6, 131.4, 131.9, 134.3, 135.3, 137.0, 141.5, 143.7, 171.3, 180.3, 180.8. ESI-MS *m*/*z* 406 (M + H)^+^. Anal.Calcd for C_16_H_11_NO_4_S_2_: C, 55.64; H, 3.21; N, 4.06. Found: C, 55.51; H, 3.09; N, 4.19.

### 3.4. Crystal Structure Determination of 11-Phenethyl-3,11-dihydro-2H-benzo[6,7]thiochromeno[2,3-d]thiazole-2,5,10-trione Dimethylaminoformamide Hemisolvate (***3.6***·1/2DMF)

Compound **3.6** was recrystallized from DMF by slow evaporation at room temperature.

*Crystal data of compound **3.6*** C_22_H_15_NO_3_S_2_, 0.5(C_3_H_7_NO), *M*_r_ = 442.02, monoclinic, space group *C*2/*c*, *a* = 19.9599(5), *b* = 8.08330(10), *c* = 26.8363(6) Å, *β* = 111.649(3)°, *V* = 4024.40(16) Å^3^, *Z* = 8 (Z’ = 1), *D*_calc_ = 1.459 g/cm3, *μ* = 2.661 mm−1, *T* = 130.0(1) K.

*Data collection of compound **3.6.*** A brown lath crystal (DMF) of 0.45 × 0.12 × 0.08 mm was used to record 9137 (Cu *K*α-radiation, *θ*_max_ = 76.22°) intensities on a Rigaku SuperNova Dual Atlas diffractometer [56] using mirror monochromatized Cu *K*α-radiation from a high-flux microfocus source (*λ* = 1.54184 Å). Accurate unit cell parameters were determined by least-squares techniques from the *θ* values of 7380 reflections, *θ* range 4.42–75.95°. The data were corrected for Lorentz, polarization, and for absorption effects [56]. The 4126 total unique reflections (*R*_int_ = 0.0146) were used for structure determination.

*Structure solution and refinement of compound **3.6.*** The structure was solved by a dual space algorithm (SHELXT) [57] and refined against *F*^2^ for all data (SHELXL) [58]. The position of the H atom bonded to N atom was obtained from the difference Fourier map and was freely refined. The remaining H atoms were positioned geometrically and were refined within the riding model approximation: C−H = 0.99 Å (CH_2_), 1.00 Å (C*sp*^3^H), 0.95 Å (C*sp*^2^H) and *U*_iso_(H) = 1.2*U*_eq_(C). Final refinement converged with *R* = 0.0584 (for 4079 data with *F*^2^ > 4*σ*(*F*^2^), *wR* = 0.1319 (on *F*^2^ for all data), and *S* = 1.144 (on *F*^2^ for all data). The largest difference peak and hole was 0.378 and −0.421 eÅ^3^. The solvent masks procedure implemented in OLEX2 [59] was employed to remove disordered solvent molecules that could not be reliably modeled. The solvent radius was set to 1.2 Å; calculated total potential solvent-accessible void volume and electron counts per unit-cell were: 484 Å^3^ and 148.

The molecular illustration was drawn using ORTEP-3 for Windows [60]. Software used to prepare material for publication was OLEX2 [59] and PLATON [61].

The supplementary crystallographic data were deposited at the Cambridge Crystallographic Data Centre (CCDC), 12 Union Road, Cambridge, CB2 1EZ (UK) [phone, (+44) 1223/336-408; fax, (+44) 1223/336-033; e-mail, deposit@ccdc.cam.ac.uk; World Wide Web, http://www.ccdc.cam.ac.uk, accessed on 2 October 2022 (deposition no. CCDC 2210721)].

### 3.5. Cytotoxic Activity against Malignant Human Tumor Cells According to the DTP NCI Protocol

Anticancer in vitro assay was performed on the human tumor cell lines panel derived from nine neoplastic diseases by the protocol of the Drug Evaluation Branch, National Cancer Institute, Bethesda, MD, USA [44,45,46]. Tested compounds were added to the culture at a single concentration (10^−5^ M), and the cultures were incubated for 48 h. Endpoint determinations were made with a protein binding dye, sulforhodamine B. Results for each tested compound were reported as the GP% of the treated cells compared to untreated control cells. GP% was spectrophotometrically evaluated vs. controls not treated with test agents.

### 3.6. MTT Cell Viability Assay

MTT assay was used to examine the viability of tumor and pseudo-normal cells after their treatment with studied thiopyrano[2,3-*d*]thiazole derivatives and doxorubicin (Actavis S.R.L., Bucharest, Romania). Cells were seeded in 96-well plates at a density of 3–5 × 103. After 24 h, cells were treated with compound **3.10**, 1,4-NQ (0.1–100 µM), and doxorubicin (0.1–100 µM). After incubation for 72 h, MTT reagent (Sigma-Aldrich, St. Louis, MO, USA) was added to each well, according to the Sigma-Aldrich protocol. An absorbance Reader BioTek ELx800 (BioTek Instruments, Inc., Winooski, VT, USA) was used to measure the reaction product.

### 3.7. Reduced Glutathione (GSH) Level Assay

In model experiments, 1 mM of GSH and 1 mM of compounds in 0.1 M phosphate buffer (pH 7.4) were incubated for 1 h at 37 °C, and then the level of GSH was determined spectrophotometrically at 412 nm based on the reduction of 5,5′-dithio-bis(2-nitrobenzoic acid) to form the yellow derivative 5′-thio-2-nitrobenzoic acid. Oxidized glutathione GSSG in samples was reduced to GSH with sodium borohydride [62].

### 3.8. Spectroscopic DNA Interaction Assay

A spectroscopic DNA interaction study was performed as previously described [63]. salmon sperm DNA (Sigma-Aldrich, USA) was diluted in Milli-Q water at 4 °C for 24 h at 1.65 mg/mL. Tested compounds were dissolved in acetone. After 1 h of incubation of DNA and compound, KMnO4 was added to a final concentration of 0.3 mM and the absorbance at 405 nm was measured (Absorbance Reader BioTek ELx800) (BioTek Instruments, Inc., Winooski, VT, USA) in different periods up to 3 h. Appropriate controls of DNA alone and compound alone were included, and these Abs values were subtracted from the test sample to provide the net change in absorbance (NetAbs). DNA-binding compounds were defined as such where the net change in absorbance was >0.05 or ˂−0.05, and DNA non-binding compounds ranged from 0.05 to −0.05.

### 3.9. DNA/Methyl Green Colorimetric Assay

The capacity of the tested compounds to intercalate into salmon sperm DNA was determined using the methyl green assay. Briefly, salmon sperm DNA (10 mg/mL) was incubated for 1 h at 37 °C with 15 µL of methyl green solution (1 mg/mL in H_2_O). The compounds were added at concentrations 1 and 10 µM/mL and incubated at 37 °C in the dark for 2 h. Reduction of the absorbance of methyl green at 642 nm induced by the test compounds was measured with a multiplate reader, Plate Reader BioTek Lx80 ( BioTek Instruments, Inc., Winooski, VT, USA). Doxorubicin, a well-known intercalating agent, was used as a positive control.

### 3.10. DNA Extraction and Gel Electrophoresis

DNA extraction and gel electrophoresis were performed as described by Herrmann and others. Jurkat cells were collected by centrifugation; lysed in a lysis buffer (1% NP-40 in 20 mM EDTA, 50 mM Tris-HCl, pH 7.5; 10 µL per 10 6 cells, minimum 50 µL). After centrifugation for 5 min at 1600× *g*, the supernatant was collected and the extraction was repeated with the same amount of lysis buffer. Supernatants were brought to 1% SDS and treated for 2 h with RNase A (final concentration 5 µg/mL) at 56 °C. Then, proteinase K was added (final concentration 2.5, µg/mL) and incubated for 2 h at 37 °C. After adding 1/2 volume of 10 M ammonium acetate, the DNA was precipitated with 2.5 vol. Ethanol, dissolved in gel loading buffer, and separated by electrophoresis in 1% agarose gels containing Ethidium bromide (at 70 V) [64].

### 3.11. The Fluorescence Microscopy of Cells

The KB3-1 cells were seeded in 24-well plates at 5 × 105/mL and then treated for an additional 24 h with compound **3.10** (1 μM) and doxorubicin (1 μM). Cells were stained with 0.2–0.5 µg/mL of Hoechst-33342 and incubated for 20–30 min before the cell examination. A Zeiss fluorescent microscope (Carl Zeiss, Jena, Germany), AxioImager A1 camera (at 400× magnification), and AxioVision image analysis software Release 4.6.3.0 for Carl Zeis microscopy (Imaging Associates Ltd., Cork, Ireland, UK) were used for cells examination. All microphotographs were additionally analyzed using ImagePro7 software (Media Cybernetics, Rockville, MD, USA) [65].

### 3.12. Statistical Data Analysis

The obtained results were analyzed and illustrated with GraphPad Prism (version 8.0.1; GraphPad Software, San Diego, CA, USA). The data were presented as the mean (M) ± standard deviation (SD), *n* = 3–4. Statistical analyses were performed using two-way ANOVA with Dunnett multiple comparisons test. A *p*-value of <0.05 was considered statistically significant.

## 4. Conclusions

This study developed an efficient method for the synthesis of thiopyrano[2,3-*d*]thiazoles containing a naphthoquinone moiety via *hetero*-Diels-Alder reaction using 5-alkyl/arylallylidene-4-thioxo-2-thiazolidinones and 1,4-naphthoquinone. The synthesized compounds were assessed for their antitumor properties according to the DTP NCI protocol. Two synthesized compounds were tested and displayed moderate antitumor activity against leukemia, non-small cell lung cancer, ovarian, breast, prostate cancer, and melanoma cell lines. The 11-phenethyl-3,11-dihydro-2*H*-benzo[6,7]thiochromeno[2,3-*d*]thiazole-2,5,10-trione (**3.6**) displayed prominent cytotoxicity effects on leukemia (Jurkat, THP-1), epidermoid (KB3-1, KBC-1), colon (HCT116wt, HCT116 p53-/-), breast (MCF-7), and carcinoma cells. The p53 status of colon carcinoma cells influenced the anti-tumor effects of the studied 3,5,10,11-tetrahydro-2*H*-benzo[6,7]thiochromeno[2,3-*d*][1,3]thiazole-2,5,10-trione. We suggest a p53-dependent mode of action for **3.6** towards colon cancer cells. The **3.6** derivative possessed moderate toxicity towards HaCaT, J774.2 cell lines, and isolated normal human lymphocytes. It induced pro-apoptotic cytomorphological changes (chromatin condensation and membrane blabbing) and mitotic catastrophe in treated KB3-1 cells. Compound **3.6** also induced a necrotic death of KB3-1 cells and interacted with DNA. The obtained data revealed the necessity for further investigations among these derivatives in modern anticancer drug therapy.

## Data Availability

Not applicable.

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
