# Peer review of "1,4-Naphthoquinone Motif in the Synthesis of New Thiopyrano[2,3-*d*]thiazoles as Potential Biologically Active Compounds"

_molecules, 2022, doi:10.3390/molecules27217575_

Round 1
Reviewer 1 Report
Dear Authors
The research article under title “1,4-Naphthoquinone Motif in the Synthesis of New Thiopyrano[2,3-d]thiazoles as Potential Biologically Active Compounds” discussed some of the newly synthesized compounds 3,5,10,11-tetrahydro-2H-benzo[6,7]thiochromeno[2,3-d][1,3]thiazole-2,5,10-triones. The structures of newly synthesized compounds were established by spectral data and a single-crystal X-ray diffraction analysis. Compound 11-Phenethyl-3,11-dihydro-2H-benzo[6,7]thiochromeno[2,3-d]thiazole-2,5,10-trione 3.6 showed pronounced cytotoxic effect on leukemia (Jurkat, THP-1) epidermoid (KB3-1, KBC-1), and colon 34 (HCT116wt, HCT116 p53-/-) cell lines. The cytotoxic action of 3.6 on p53 deficient colon carcinoma cells was two times weaker than on HCT116wt. However, there are some discrepancies regarding to the characterization the results and discussion and the part of the experimental section, as follows.
1- Line 32, write the chemical name first of 3.6 in the abstract.
2- Line 95, what mean by ethanol, the catalyst. Do you mean ethanol/catalyst.
3- Line 105, write the chemical name of EDDA.
4- Line 116, Letter d corrected to chemical shift d.
5- Line 129, correct scheme into Scheme, also inn line 131.
6- Line 209, correct 9,19, 8,47 into 9.19, 8.47 and also the other numbers in Table 4 in line 236.
7- Line 359, check the J coupling constant (7.9 MHz), due to you have 400 MHz.
8- Line 381, check the data of 1H NMR for compound 3.4, due to the number of proton not matched the molecular formula of it.
9- Also the author check the 1H MNR for compounds 3.5and 3.6 due to the number of protons not compatible with molecular formula of the compounds.
10- Compound 3.6 has melting point above 350 oC, the authors check again the solvent of crystallization.
Author Response
Dear Reviewer!
We would like to thank You for the revision and constructive comments that helped significantly improve the manuscript. Your suggestions have been incorporated in the revised manuscript (yellow highlight).
We would like to comment the main points.
“Line 95, what mean by ethanol, the catalyst. Do you mean ethanol/catalyst.”
The reaction was accomplished in an ethanol medium and in the presence of ethylenediammonium diacetate as a catalyst.
Following Your comments, we also clarified the data of 1H NMR and molecular formulas for synthesized compounds, inserted the name of EDDA, and clarified the melting point for compound 3.6. In addition, we carefully revised the manuscript language and hope the current version will be acceptable for publication.
Reviewer 2 Report
In this manuscript, Lesyk and coworkers synthesized a novel series of thiopyrano[2,3-d]thiazoles derivatives and evaluated their primary anticancer activity, and cytotoxicity. They also studied the mechanism of action through different DNA intercalation in vitro studies. Synthesis of core structure was nicely executed through hetero Diels-Alder reaction of 1,-naphthoquinones and 2-thiazolidinones. Anticancer screening data for compounds 3.5 and 3.6 is promising and compound 3.6 has an excellent cytotoxicity profile against different cancer cell lines. Considering the utility and future prospect of these novel compounds, this manuscript should be considered for publication in Molecules after addressing following comments.
1. Although introduction of this manuscript is very extensive, reference regarding the biological activity of thiopyrano[2,3-d]thiazoles is missing. A few references regarding anticancer activity of thiopyrano[2,3-d]thiazoles should be added in the introduction section; for example: Sci. Pharm. 2012, 80, 509–529; Chem. Pharm. Bull. 2015, 63, 495–503 and ref 37.
2. What is the reason behind choosing compounds 3.5 and 3.6 over compounds 3.1-3.4 for anti-cancer activity study? A rationale should be added.
3. Figure 2 is missing reference in the main text.
4. In Line 112, C-11 position should be corrected into C-16 position.
5. In Scheme 1, R3-group notation for compound 2.5 and 3.5 should be corrected. Considering the cyclopentane ring in the structure, R3+R3 should be equal to (CH2)4.
6. Table numbering should be checked in the main text as well as in the table headings. Numbering has been started from number 3.
7. In Table 4, IC50 values need to be checked for HCT116wt, HCT116 p53-/- and MCF-7 cell lines. Comma and dot have been exchanged here. Similar concern for line 209 and 210.
8. The viability curve of compound 3.6 against Jurkat, THP-1, MCF-7 and K562 cell lines are missing in the manuscript. These curves should be added in the published version.
9. In line 386: molecular formula for 3.4 should be corrected. Similar concern for compound 3.5 (line 392) and compound 3.6 (line 404).
10. In supporting info, a clear 1H NMR spectra (high resolution, low signal to noise ratio) for compound 3.5 should be reported.
Author Response
Dear Reviewer!
We would like to thank You for the revision and constructive comments that helped significantly improve the manuscript. Your suggestions have been incorporated in the revised manuscript (yellow highlight).
We would like to comment the main points.
“Although introduction of this manuscript is very extensive, reference regarding the biological activity of thiopyrano[2,3-d]thiazoles is missing. A few references regarding anticancer activity of thiopyrano[2,3-d]thiazoles should be added in the introduction section; for example: Sci. Pharm. 2012, 80, 509–529; Chem. Pharm. Bull. 2015, 63, 495–503 and ref 37”
According to Your suggestions, we added the references regarding the biological activity of thiopyrano[2,3-d]thiazoles in the current manuscript.
“What is the reason behind choosing compounds 3.5 and 3.6 over compounds 3.1-3.4 for anti-cancer activity study? A rationale should be added.”
All synthesized compounds were submitted to the National Cancer Institute (NCI), but only two were selected for biological activity screening. The process of selecting compounds is carried out by the NCI small molecule compound collection's selection guidelines, and the information mentioned was added in the current manuscript.
The viability curve of compound 3.6 against Jurkat, THP-1, MCF-7 and K562 cell lines are missing in the manuscript. These curves should be added in the published version “
Compound 3.6 was the most active in tumor cells compared to other derivatives. It possessed moderate toxicity on pseudo-normal ones. Thus, compound 3.6 was chosen for further experimental research in vitro.
“In supporting info, a clear 1H NMR spectra (high resolution, low signal to noise ratio) for compound 3.5 should be reported.”
Following Your suggestions, we improved the quality of 1H NMR spectra for compound 3.5 in supplementary data.
In addition, following Your comments, we also clarified the molecular formulas for compounds 3.4-3.6, the numbering and references of figures and tables in the current manuscript, carefully revised the manuscript language, and hope the current version will be acceptable for publication.
Kind regards,
Authors